# Ultrathin silica-tiling on living cells for chemobiotic catalysis

Jeongsang Oh[1,2,4], Nitee Kumari[1,2,4], Dayeong Kim[1,2], Amit Kumar ®[1,2,5] ✉ & In Su Lee ®[1,2,3,5] ✉

Harnessing the power of cell biocatalysis for sustainable chemical synthesis requires rational integration of living cells with the modern synthetic catalysts. Here, we develop silica-tiling strategy that constructs a hierarchical, inorganic, protocellular confined nanospace around the individual living cell to accommodate molecularly accessible abiotic catalytic sites. This empowers the living microorganisms for new-to-nature chemical synthesis without compromising the cellular regenerative process. Yeast cell, a widely used biocatalyst, is upgraded via highly controlled self-assembly of 2D-bilayer silica-based catalytic modules on cell surfaces, opening the avenues for diverse chemobiotic reactions. For example, combining [AuPt]-catalyzed NADH regeneration, light-induced [Pd]-catalyzed C-C cross-coupling or lipase-catalyzed esterification reactions—with the natural ketoreductase activity inside yeast cell. The conformal silica bilayer provides protection while allowing proximity to catalytic sites and preserving natural cell viability and proliferation. These living nanobiohybrids offer to bridge cell's natural biocatalytic capabilities with customizable heterogeneous metal catalysis, enabling programmable reaction sequences for sustainable chemical synthesis.

Empowering living microorganisms and cells with the capability to perform new-to-nature metal-catalyzed reactions can hugely broaden the scope of artificial "chemo-bio" catalysis in diverse fields including sustainable energy conversion, fine-chemicals synthesis, antimicrobials, therapeutic probiotics, biomedicine, cell-therapy and nanobiotechnology[1–6]. For example, direct integration of semiconductor nanocrystals (NCs) with living cell (yeast, bacteria and plants) by bio-precipitation or polyphenol-polymeric glue as well as intracellular localization are useful in harnessing photo-generated electrons in solar-to-chemical synthesis[7–9]. Few cell-encapsulation strategies involving metal-organic frameworks, polydopamine, polymer and coordination polymer coatings can localize large-size enzymes within membrane proximity[10–13]. Direct biocompatible cell-silicification, through tuning the structure, thickness and conformity of the sol-gel silica coatings, offers the potential to manipulate the

behaviors of encapsulated cells, inducing stress-resistant states in metabolically viable but non-culturable states for various applications[14–18]. Other strategies involving layer-by-layer electrostatic deposition of polymers, titania and silica are highly effective for the long-term cytoprotection but they also subside the normal metabolism and cell-division processes due to their thick (>100 nm) and static shell-like enclosure[18–24]. Unfortunately, besides the random nanoparticle (NP) deposition or entrapment of the whole-cells inside bulk support materials (such as polymers, hydrogels, and silica)—generalizable strategies to engineer living cells with the diverse catalytic modalities have not been developed due to several challenges: (i) catalytically useful small-size metal NCs (<5 nm), complexes or enzymes are prone to deactivate in biomedia (ii) facile molecular diffusion and interactions with the active-site are often obstructed by the crowded microenvironment and (iii) the interfacial delicate and

[1]Creative Research Initiative Center for Nanospace-confined Chemical Reactions (NCCR), Pohang University of Science and Technology (POSTECH), Pohang 37673, Korea. [2]Department of Chemistry, Pohang University of Science and Technology (POSTECH), Pohang 37673, Korea. [3]Institute for Convergence Research and Education in Advanced Technology (I-CREATE), Yonsei University, Seoul 03722, Korea. [4]These authors contributed equally: Jeongsang Oh, Nitee Kumari. [5]These authors jointly supervised this work: Amit Kumar, In Su Lee. ✉e-mail: amitkumar@postech.ac.kr; insulee97@postech.ac.kr

dynamic living cells must experience minimal stress and toxicity; thereby, random clustering of foreign reactive materials directly on to the cell membranes should be avoided[25–27]. Already, few multi-component architectures of porous silica-based nanoreactors have endowed broad range of reactivity profiles for intra/extra-cellular reactions[28–32]. However, the location and fate of those spherical shape nanostructures inside/on cell surface is difficult to control due to their random clustering on membrane or inside endosomes. Catalytic performance of encapsulated metals sensitively depend on their spatial localization and facile mass-transport which are highly challenging to engineer and sustain[33–35]. Simultaneously, living cells should maintain inherent metabolic flux, proliferation and recyclability. Also, biocompatible interfacing chemistry should involve least number of steps and minimal use of complex reagents to avoid any cytotoxicity[36–39]. Therefore, rational integration of living cells with versatile metal-based catalysts, is highly challenging to achieve. For example, baker's yeast (*Saccharomyces cerevisiae*) microbial cell, is the most used whole-cell biocatalyst, performing various types of chemo-, regio- and stereo-selective reactions[40]. However, the natural cells are limited for only few

types of chemical transformations mainly depending on their oxidoreductase activity and difficult to be directly implemented in one-pot multistep chemical reaction to synthesize complex molecular structures[41].

In this work, we construct a hierarchical inorganic protocellular confined nanospace around the individual living yeast cell, like a functional bionic jacket. The key strategy involves the use of customizable catalytic compartment modules−2D-bilayer porous silica nano-tiles (2D-SiNTs), hosting diverse metal NCs (e.g. Au, Pd, Pt etc.) of choice or non-native enzymes inside the bilayer nanospace−laterally close-packing as the result of directed self-assembly, evolving in to a conformal, flexible and ultrathin biocompatible enclosure on to the delicate individual cell to perform new-to-nature chemobiotic reaction sequence (Fig. 1). Such biocompatible silica-tiling process has negligible influence on the viability, metabolic status and proliferation of encapsulated living cell. Our strategy allows to harmoniously combine choice of abiotic catalytic reaction step−for example, [AuPt]-catalyzed nicotinamide adenine dinucleotide (NADH) regeneration or localized surface plasmon resonance (LSPR)-induced [Pd]-catalyzed C-C cross-

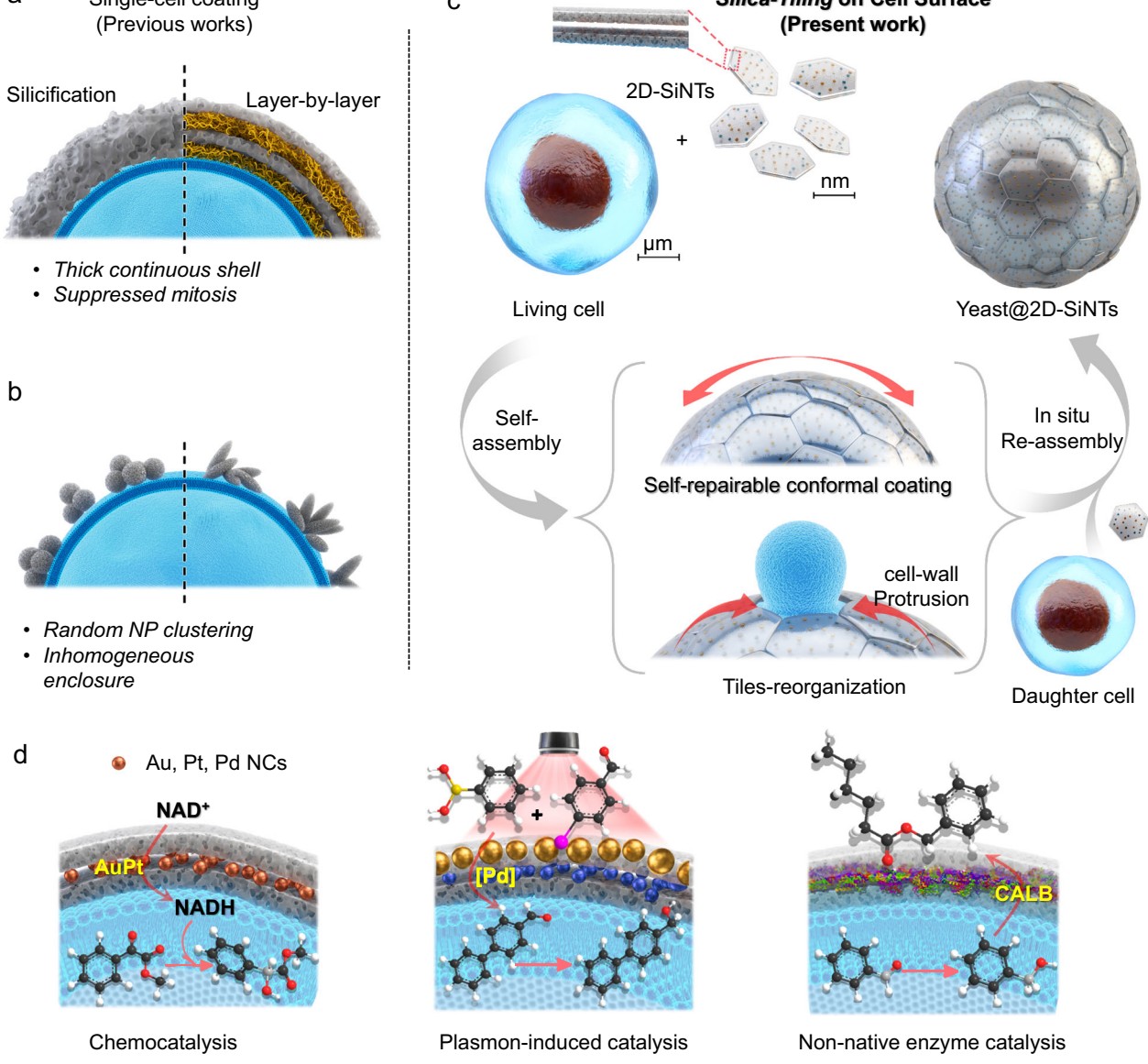

**Fig. 1 | Conceptual representation of Silica-Tiling process on living cell for chemobiotic catalysis. a, b** Previously known single-cell coating methods; **c** Schematic for silica-tiling and its self-adaptive behavior during cell division; **d** Customizable synthetic catalytic modalities based on silica-tiling over living cell surface to combine with cell's own biocatalysis.

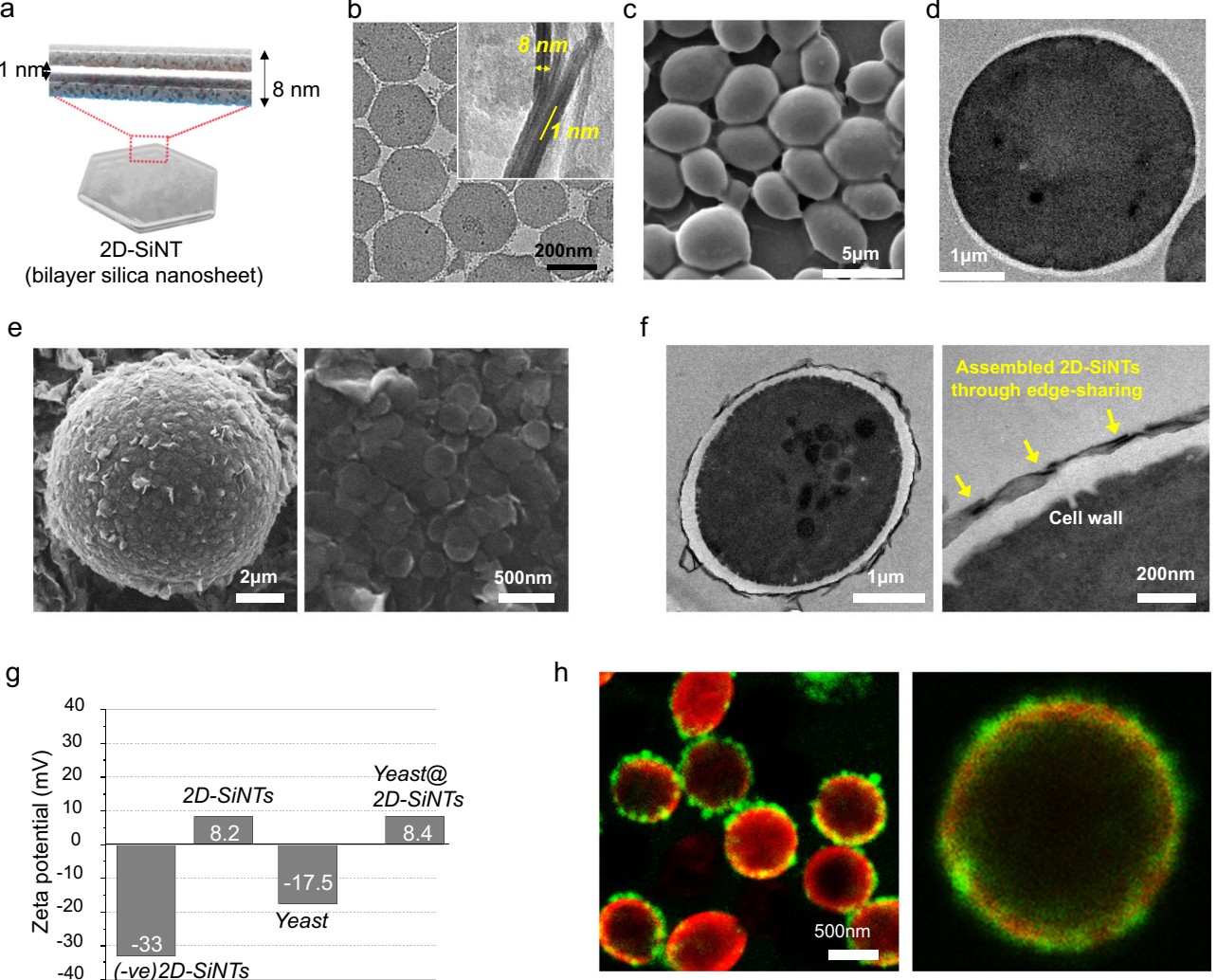

**Fig. 2 | Characterization of Yeast@2D-SiNTs. a, b** Pictorial representation and TEM images of 2D-SiNTs, inset shows the standing sheet TEM image. **c, d** SEM (whole cell) and Bio-TEM image (cross-sectioned) of native yeast cells. **e, f** SEM (whole cell) and Bio-TEM image (cross-sectioned) of Yeast@2D-SiNTs. **g** Average zeta potentials of 2D-SiNTs, yeast cell and yeast@2D-SiNTs. Source data are provided as a Source Data file. **h** CLSM fluorescence images of Yeast@2D-SiNTs yeast cell membrane is labeled with red and 2D-SiNTs are labeled with green fluorophores. All micrographic images (**b–f, h**) represent the results of at least 3 independent experiments.

coupling or Lipase-catalyzed esterification reactions–with the natural ketoreductase activity inside yeast cell in order to efficiently perform "one-pot-two-step" chemobiotic reaction sequence.

## Results and discussion
### Silica-tiling strategy for single cell encapsulation
First, we synthesized positively charged 2D-SiNTs (200 nm lateral size, 8 nm thickness, + 8.2 mV zeta potential) following a known sol-gel chemistry based protocol (details in Supplementary Information, Supplementary Fig. 1, Fig. 2a, b)[42]. 2D-SiNT has a silica bilayer morphology with customizable interior nanospace (~1 nm) where different catalytic metals can be selectively functionalized. In a typical silica-tiling procedure, 2D-SiNTs (0.3 mg in 5 mL DI) were mixed with the freshly cultured yeast cells ($4 \times 10^8$ cells mL$^{-1}$) for 5 min. Silica coated yeast cells (Yeast@2D-SiNT) were collected by centrifugation (2000 rpm, 3 min) and washed with DI. Scanning electron microscopy (SEM) showed conformal coating of individual cell by the uniform closely packed self-assembled 2D-SiNT units intimately, adhering on to the cell-wall as hexagonal patches through the large-face, plausibly, due to the higher positive charge density on the face in comparison to the edges (Fig. 2c–e). Noticeably, after silica-tiling, smooth surface of

original yeast cell was converted to slightly rough flaky morphology due to the presence of self-assembled 2D-SiNTs (Fig. 2c and Fig. 2e). Transmission electron microscopy (TEM) of fixed and cross-sectioned cell revealed almost monolayer coating (dark contrast) of individual cell where 2D-SiNTs support each other through slight edge-overlapping and sharing (<10 nm thick) closely interfacing with the light-contrasted thicker cell wall (Fig. 2f). Remarkably, no adverse effect on cell wall integrity was noticed due to the coating treatment. After silica modification on cell surface, originally negative zeta potential (−17.5 mV) was switched to +8.4, indicating the full coverage of yeast surface with positively charged 2D-SiNTs (Fig. 2g). Upon switching the charge of 2D-SiNTs to negative, 2D-SiNTs were de-attached from the cell surface, validating the reversible and dynamic electrostatic force driving the tiling process (Supplementary Fig. 2). High resolution confocal laser scanning microscopy (CLSM) of Yeast@2D-SiNT clearly visualized the ultrathin corona of fluorescein-labeled 2D-SiNT layer (green) uniformly interfacing with the cell wall (labeled red) (Fig. 2h). Also, CLSM-based 3D Z-scanning of the whole cell revealed homogeneous enclosure of cell by 2D-SiNTs only on the cell surface (Supplementary Movie 1). Heterogeneity of green fluorescence (tiling) in high-mag CLSM images may be due to different

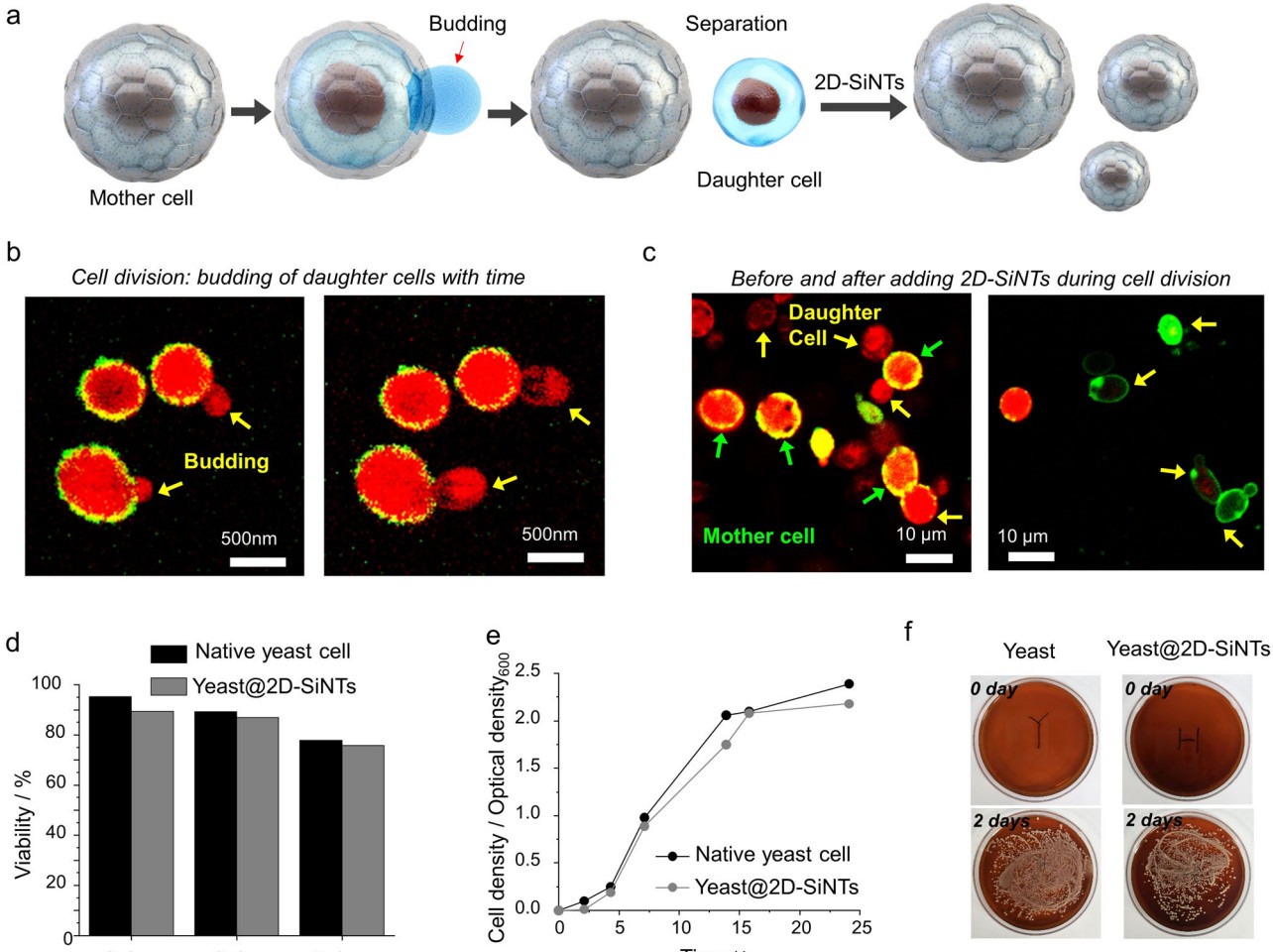

**Fig. 3 | Cell-division study of Yeast@2D-SiNTs. a** Schematic representation of cell-division process of Yeast@2D-SiNTs. **b** Time-dependent live cell imaging demonstrating the cell division process, yellow arrows showing emergence of daughter cells by protruding through 2D-SiNTs on Yeast@2D-SiNTs surface. **c** Fluorescence images before (left) and after (right) adding additional 2D-SiNTs to modify the surface of daughter cells. **d** Comparative cell viabilities measured at different incubation periods measured by flow cytometry analysis. **e** UV-vis spectrophotometry based quantification of increase in number of cells in suspensions with time. **f** Digital camera images of cell-culture on YPD agar modified petri-dishes at after 7 days showing the emergence of comparable cell colonies in the case of Yeast and Yeast@2D-SiNTs. All micrographic images (**b**, **c**) represent the results of at least 3 independent experiments. Source data (**d**, **e**) are provided as a Source Data file.

reasons: (i) ultrathin tiling provides much weaker and fluctuating fluorescence signals as compared to the interfacial thick cell-wall; (ii) slight movements or adjustments of nanoscale silica tiles on living and slightly mobile cells (non-fixed) can't be fully suppressed during measurements. These heterogeneity factors may be broadly resolved and visualized in high-mag live-cell CLSM imaging. However, low-mag CLSM images (Supplementary Fig. 3) of crowded cells showed reasonably homogeneous presence of green 2D-SiNTs fully coated on every individual yeast cell.

**Silica-tiles during cell-division process**

To investigate the effect of silica-tiling on encapsulated cell's natural processes, we conducted CLSM-based live-cell imaging study on the real-time cell division process of Yeast@2D-SiNTs (Fig. 3a, Supplementary Movie 2). As shown in Fig. 3b, budding daughter-cell protrude out of thin silica enclosure without carrying away any 2D-SiNT on their newly evolving surface and naturally separate from the mother cell, turning in to fresh native cell. It is to be noted that after cell division, there was minimal change of silica enclosure on the mother cell, still showing uniform silica enclosure, as evident from the homogeneous peripheral green fluorescence intensity on cell surface. The emergence of the daughter cell may occur through pinching off the 2D-SiNT shared edges, opening a sufficiently larger size gate to facilitate the

outwardly protruding daughter cell-growth. Conveniently, new-born daughter cells can be further modified by re-introducing 2D-SiNTs in to the cell culture, again forming uniform coating around individual daughter cell (Fig. 3c). Such unique in situ dynamic "associative-dissociative" nature of ultrathin silica enclosure is distinct from the conventional rigid and thick coatings reported so far. Remarkably, cells maintain their normal biological processes while 2D-SiNTs remain adhered to the cell wall like their own integral component. Ultrathin flat sheet-like morphology of 2D-SiNTs had crucial effect on the self-assembly behavior and in turn homogeneous skin-like coating on the cell surface.

Next, we estimated the cell viability by live/dead cell assay and flow cytometry at different incubation periods (0, 3, 7 days) verifying the comparative cell viability of Yeast@2D-SiNT and unmodified yeast cells to be similar, which ensured the unaffected membrane integrity and internal esterase activity of the cells upon silica-tiling (Fig. 3d, Supplementary Fig. 4). In addition, a UV-vis based quantification of cell density in the liquid cell growth media showed a typical growth kinetics, cell numbers increased due to the natural division process in both the cases (Fig. 3e). In corroboration, Yeast@2D-SiNTs generated typically grown cell-colonies on solid YPD agar surface in a petri-dish, similar to the native yeast growth (Fig. 3f, Supplementary Fig. 5). These results indicated that the silica-tiling process was biocompatible and

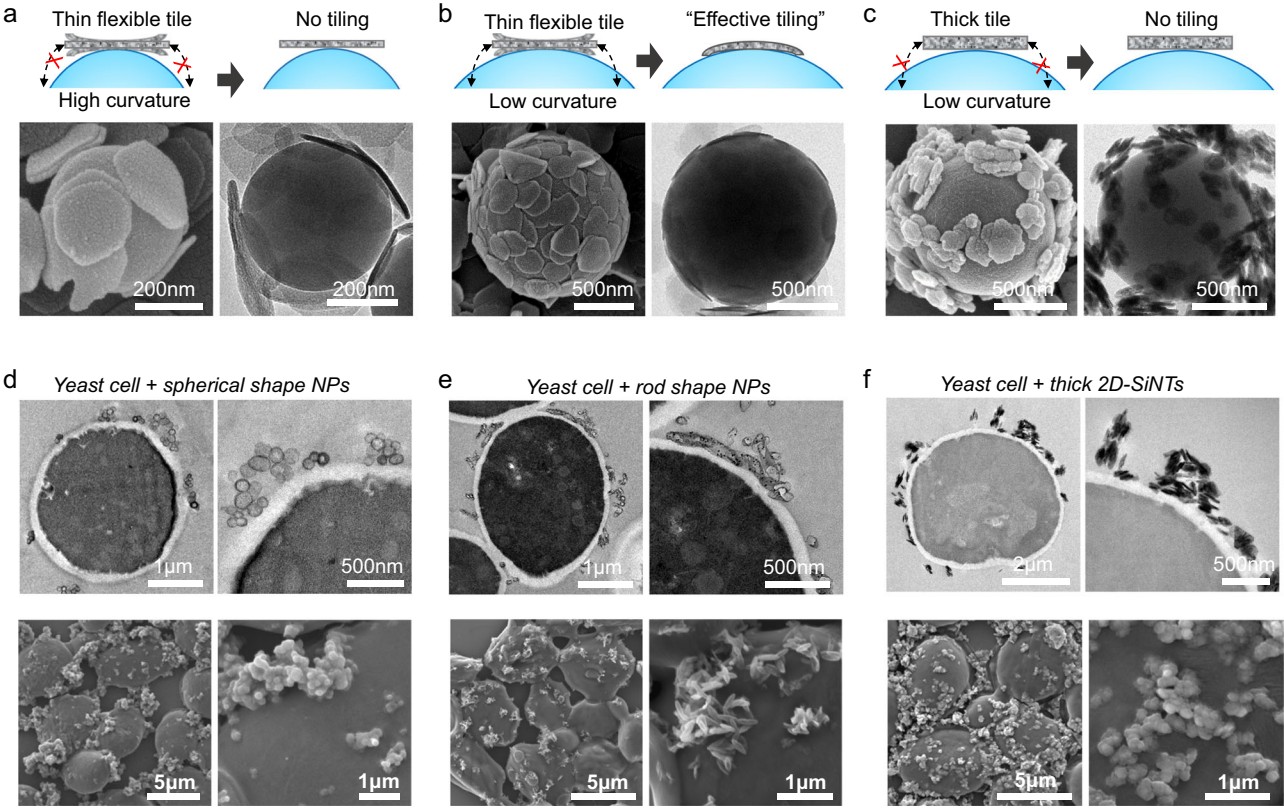

**Fig. 4 | Effect of different NP shapes on silica-tiling. a–c** Mechanistic depiction (top), SEM image (down left) and TEM images (down right) of thin / flexible and thick 2D-SiNTs assembly on polystyrene microspheres of different curvatures; thin and flexible 2D-SiNTs show conformal monolayer coverage on optimally curved surface. **d–f** Cross-sectioned Bio-TEM images (top) and SEM images (down) of different silica NP shapes assembled on yeast surface, showing random NP clustering, unlike thin/flexible 2D-SiNTs. All micrographic images (**a–f**) represent the results of at least 3 independent experiments.

had minimal influence on the natural cell growth processes by optimally maintaining the flow of nutrients and natural cell division processes across ultrathin silica layer.

**Mechanistic study of silica-tiling process**

In order to further investigate the critical mechanistic factors responsible for a compact and conformal coating through self-assembly process, we conducted the silica-tiling on negatively charged polystyrene microspheres (PS-MS) of well-defined sizes and surface composition (Fig. 4a–c). In first control study, we assembled 2D-SiNTs on PS-MS of different sizes (500 and 1000 nm). Due to the change in size-dependent PS-MS's surface curvature, 2D-SiNTs demonstrated compact and conformal assembly only on the PS-MS of 1000 nm size, which provided optimum interfacial contact area for effective electrostatic adhesion, mostly as monolayer (Fig. 4b). In another control study, 2D-SiNTs with increased thickness (ca. 20 nm), showed random clustering around the PS-MS of different surface curvatures (Fig. 4c). Interestingly, change in interfacial contact area between thick 2D-SiNT and PS-MS surface had no influence on the random assembly behavior. These studies revealed the critical role of interfacial contact area between 2D-SiNT and cell surface along with the thickness of 2D-SiNT to succeed in uniform silica-tiling. Optimally thin 2D-SiNTs are flexible enough to assume the spherical surface curvature and maximize the electrostatic adhesion by slightly bending while inter-sheet face-to-face stacking is avoided by the electrostatic repulsion so that only monolayer is continuously formed through edge-to-edge supporting around curved cell-wall. Such structural flexibility and surface chemistry is unique in highly anisotropic 2D-SiNTs which is distinct from the other shapes. Consequently, other silica NP shapes such as nano-spheres, nano-rods and thick sheets

having similar positive surface charges randomly assembled like collection of clusters in the vicinity of cell surface, as verified by the SEM and Bio-TEM images (Fig. 4d–f, Supplementary Fig. 6). Further, we have assessed the protective characteristics of the tiled cells against various stressors by estimating the cell-viabilities after exposure to different conditions (Supplementary Fig. 7). Under identical conditions, cell viabilities of native yeast cells and Yeast@2D-SiNTs were reasonably comparable or slightly better in the latter case. No dramatic influence of silica-tiling on cell viability against strong physical forces (stirring, sonication and pressure) was observed. Also, small size stressors (organic solvent, acid and radicals) appeared to interact with native Yeast and Yeast@2D-SiNTs in similar way. However, silica-tiling significantly suppressed the cell death upon exposure to the cell-wall degrading Lyticase enzyme as compared to the native yeast, possibly, due to the restricted interaction of large-size enzymes with the cell-membrane. It is to be noted that entry of few enzymes across rarely available inter-tile junctions can't be fully suppressed. Such results are expected due to the ultrathin self-assembled morphology of microporous bilayer silica nanosized units, which may provide only moderate optimal protection to further employ them in catalytic reactions. Further, we treated Yeast@2D-SiNTs with 2-(7-Nitro-2,1,3-benzoxadiazol-4-yl)-D-glucosamine (2-NBDG), a fluorescent tracer used for monitoring glucose uptake into living cells (Supplementary Fig. 8). Confocal microscopy detected emergence of intracellular green fluorescence due to the facile 2-NBDG diffusion and cellular uptake in Yeast@2D-SiNTs, similar to the native yeast cells. Whereas, membrane-impermeable dye [Dil $C_{18}$(3)] molecules (red fluorescence) accumulated on the cell wall in both cases. Considering the small molecular size (<1 nm), diffusion of 2-NBDG is consistent with the expected microporosity of amorphous silica 2D-SiNTs enclosure. Overall,

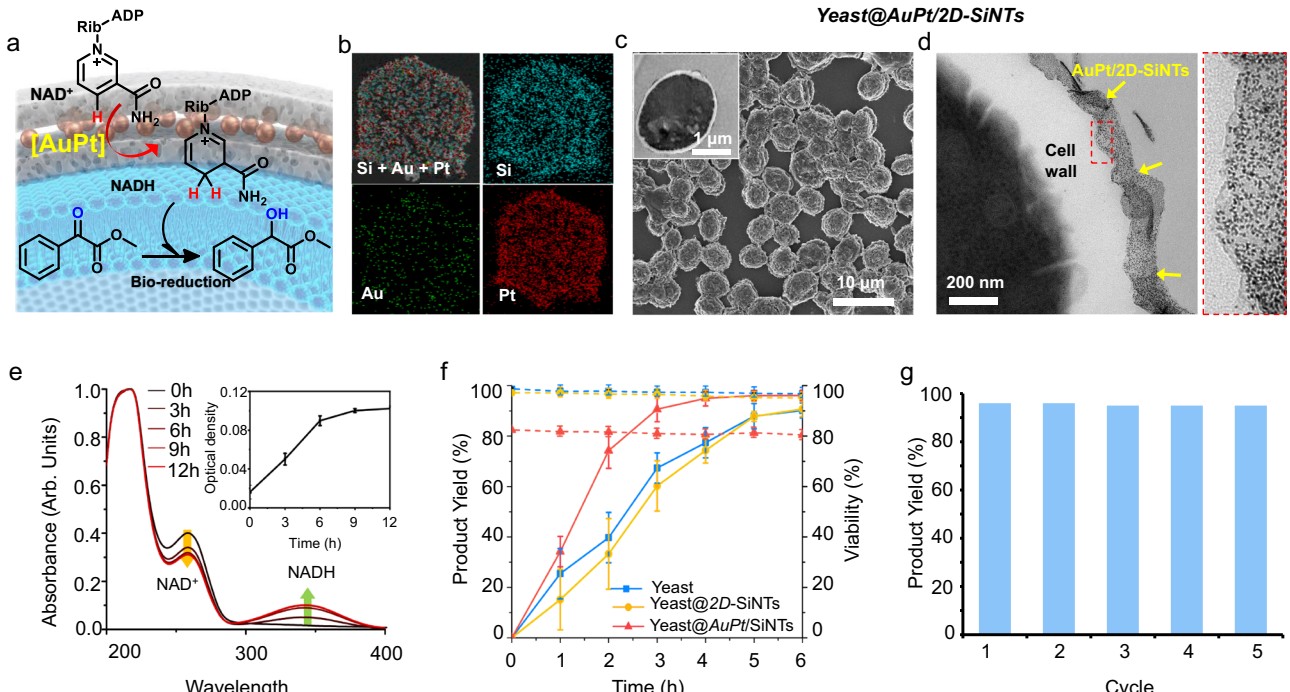

**Fig. 5 | Silica-tiling to combine AuPt-catalyzed NADH regeneration with cell's bio-reduction. a** Schematic representation. **b** STEM-EDS based elemental mapping of AuPt/2D-SiNT. **c, d** SEM and Bio-TEM images AuPt/2D-SiNTs assembled on yeast surface. **e** UV-Vis spectra of NADH regeneration by Yeast@AuPt/2D-SiNTs at different reaction times, inset shows time-dependent increase in O.D values at 360 nm. Error bars in the inset plot represent standard deviation of the mean obtained by 3 independent experiments. **f** Comparative reaction kinetics of methyl benzoylformate bio-reduction in the presence of NAD$^+$. Error bars in all plots represent standard deviation of the mean obtained by 3 independent experiments. **g** Recyclability test of Yeast@AuPt/2D-SiNTs for methyl benzoylformate reduction. All micrographic images (**b**–**d**) represent the results of at least 3 independent experiments. Source data (**e**–**g**) are provided as a Source Data file.

diffusion of small molecules for prospective catalytic reactions across silica-tiles is not restricted and is very similar to the native yeast.

## Combining abiotic NADH-regeneration and bio-reduction

Bio-catalysis, involving the oxidoreductase-reactions, depends on the sufficient availability of co-factor hydride source such as NADH[43]. For practical applications, developing metal-catalyzed abiotic regeneration of NADH is important to address the bottlenecks such as high cost, stoichiometric usage, and physical instability of NADH[44]. We planned to employ Yeast@2D-SiNTs for the [AuPt]-catalyzed abiotic NADH regeneration in order to supplement the yeast's own keto-reductase bio-reduction performance (Fig. 5a). For this, we synthesized different catalytic modules of 2D-SiNTs (designated as M/2D-SiNTs, M = AuPt) by a simple reductive metal NCs (<2 nm) deposition selectively, at the interior space of silica bilayer (details in Supplementary Information, Fig. 5b, Supplementary Fig. 9). Mixing positively charged AuPt/2D-SiNTs with the freshly cultured yeast cell suspension produced a typical ultrathin layered assembly over individual yeast cell. These catalytic metal functionalized nanobiohybrids (designated as Yeast@AuPt/2D-SiNTs) were characterized by SEM, TEM and X-ray photoelectron spectroscopy (XPS) (Fig. 5c, d, Supplementary Figs. 10–12).

In a catalytic reaction test, UV-vis spectrophotometry monitored continuous production of NADH from β-NAD$^+$ in the presence of Yeast@AuPt/2D-SiNTs over 12 h at 30 °C, as confirmed by the gradual increase in absorbance at 340 nm (due to NADH production) and correlated decrease in the absorbance at 260 nm (due to β-NAD$^+$ consumption) (Fig. 5e). Next, in a reaction mixture containing Yeast@AuPt/2D-SiNTs, reduction of ketoester (methyl benzoylformate) to the chiral hydroxyl ester (methyl 2-hydroxy-2-phyenyl acetate) in the presence of exogenous β-NAD$^+$ (as a precursor for NADH) was estimated to be in ca. 80% yield within 2 h at 30 °C (Fig. 5f,

Supplementary Fig. 13). In the control experiments, identical reactions using native yeast and Yeast@2D-SiNTs (without AuPt) were much slower, giving <40% conversions at 2 h reaction time. In another control experiment, we confirmed the similar yeast bio-reduction rate enhancement effects upon adding NADH exogenously (Supplementary Fig. 14). These results confirmed that the Yeast@AuPt/2D-SiNTs can successfully combine of abiotic NADH regeneration step to enhance the rate of yeast's bio-reduction reaction. These nanobiohybrids were easily separated from the reaction and re-used for at least 5 cycles affording consistent high yield (>95%) of chiral hydroxyl ester (Fig. 5g). To analyze the stability of hybrid catalyst, we performed Bio-TEM, ICP-AES elemental quantification and cell-viability studies on the Yeast@AuPt/2D-SiNTs isolated after reaction. Bio-TEM images of the fixed and sectioned cells revealed reasonably intact positioning and layered morphology of 2D-SiNTs on cell-membrane (Supplementary Fig. 15). Comparing the ICP-AES of the catalyst pallet (after centrifugation) and supernatant solution, before and after reaction, estimated almost no leaching of Au and Pt metals. In addition, cell viability studies on the hybrid catalyst isolated after reaction, revealed minimal loss (<5%) in living cell population. Upon increasing the Pt amounts (ca. 3, 9 and 17 wt% Pt), the active metal catalyst for NADH regeneration, in Yeast@AuPt/2D-SiNTs, exogenous NADH generation rates were enhanced (Supplementary Fig. 16). And, in response, benzoylformate bio-reduction rates were also increased in the one-pot sequential reaction. These results indicated the effective coupling of on-membrane catalytic NADH generation reaction step with intracellular bio-reduction reaction step. Lipophilic NADH molecules produced on membrane-interfacial catalytic sites inside silica tiles, are efficiently taken up by the encapsulated cells, enhancing the yield of intracellular bio-reduction step. These results also confirmed the minimal influence of silica-tiling on the natural microbial reductive activity of the encapsulated yeast, also, ultrathin amorphous silica layer facilitated

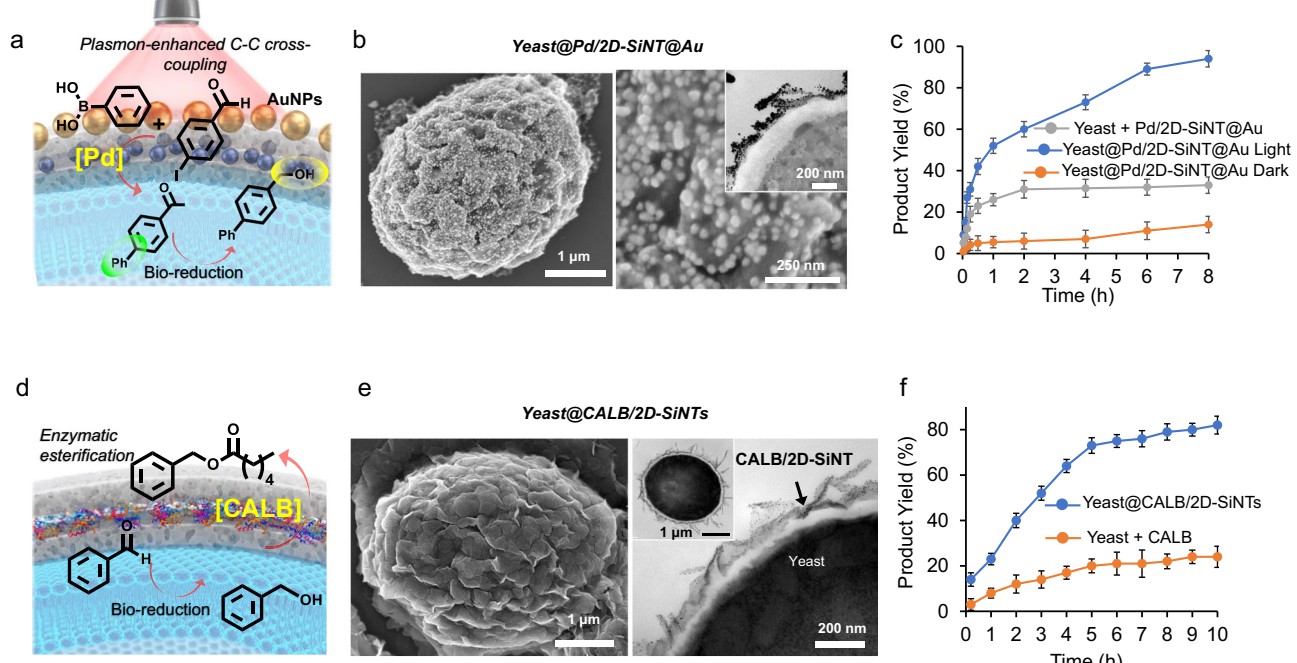

**Fig. 6 | Silica-tiling to combine different catalytic modalities with cell's bio-reduction. a** Schematic for combining plasmon-enhanced C-C cross coupling and cell's bioreduction reactions. **b** SEM images of Yeast@Pd/2D-SiNT@Au, inset shows cross-sectioned TEM image. **c** Reaction kinetics of the formation of final alcohol product under different conditions. Error bars in all plots represent standard deviation of the mean obtained by 3 independent experiments. **d** Schematic for combining cell's bioreduction and CALB-mediated esterification reactions. **e** SEM (left) and cross-sectioned Bio-TEM images of Yeast@CALB/2D-SiNT. **f** Reaction kinetics of the formation of final ester product under different conditions. Error bars in all plots represent standard deviation of the mean obtained by 3 independent experiments. All micrographic images (**b**, **e**) represent the results of at least 3 independent experiments. Source data (**c**, **f**) are provided as a Source Data file.

unobstructed flow of small size reactant and product molecules across the abundant pores and inter-sheet gaps among assembled monolayer of 2D-SiNTs. In a control experiment, yeast cells assembled with the spherical shape hollow silica NPs containing AuPt, afforded ca. 23% lower yield of benzoylformate reduction as compared to Yeast@2D-SiNTs, under identical conditions (Supplementary Fig. 17). It may be due to the intimate monolayer interfacing of AuPt/2D-SiNTs catalysts on to the cell-surface, cooperating more effectively with the cell-biocatalysis. It is to be noted that AuPt/2D-SiNTs alone (no yeast), afforded <20% yield of hydroxyl ester (Supplementary Fig. 18).

### Combining plasmonic-catalytic and bio-reduction catalysis

Synthetic catalysts seldom operate under ambient conditions, and high temperatures are usually needed to overcome the activation energy barrier. However, conventional bulk heating is not suitable for the delicate living cells. We planned to integrate plasmonic component as a plug-in to remotely promote sluggish abiotic chemical reactions by light exposure. LSPR-induced localized energy flow can be highly advantageous in promoting interfacial catalytic reactions without heating the whole reaction media[45]. We extended the silica-tiling approach by first assembling Pd/2D-SiNT modules of choice on yeast surface, followed by the electrostatic binding of pre-synthesized negatively charged citrate-capped AuNPs resulting the decoration of a plasmonic NPs over silica encapsulated yeast shell, designated as Yeast@Pd/2D-SiNT@Au (Fig. 6a, b, Supplementary Figs. 19–21). Such compact layer-by-layer nano-assembly process generated a close proximity of outer plasmonic Au with the catalytic compartment. We employed Yeast@Pd/2D-SiNT@Au for a one-pot-two-step sequential reaction: Suzuki-Miyaura coupling of 4-iodo benzaldehyde and phenyl boronic acid followed by the bio-reduction of cross-coupled product (aldehyde) to the corresponding alcohol (Fig. 6a, Supplementary Fig. 22). Dark condition resulted <20% yield of final alcohol product even after extending the reaction for >10 h (Fig. 6c). Remarkably,

exposing the reaction with 785 nm laser (0.3 W/cm²), produced >60% alcohol within 4 h, and reaction was completed giving 99% yield of alcohol in <10 h at 30 °C. In a control experiment, a physical mixture of Yeast cells and Pd/2D-SiNT@Au produced <40% yield of alcohol under identical conditions. These results indicated the crucial influence of the proximity of 'plasmonic-catalytic' components directly interfacing with the encapsulated yeast functioning as biocatalyst: efficient LSPR-induced cross-coupling reaction on the cell surface avails intermediate aldehyde in high concentrations for bio-reduction step inside yeast.

### Combining cell bio-reduction and non-native enzyme catalysis

As previously discussed, type of bio-catalysis reaction by a natural cell depends on the limited type of genetically encoded enzymetic reactions Engineering natural cells to genetically express non-native enzymes is highly challenging. Using a physical mixture of cell with isolated enzyme can risk the de-naturation of enzymes in stressful reaction conditions and uncontrolled flux of molecules between two different and isolated catalytic entities. In order to expand the scope of different enzymes-based biocatalysis, we implemented our silica-tiling strategy to locate non-native enzyme, well-preserved inside the silica bilayer nanospace, assembled over yeast cell. For proof-of-concept, we encapsulated Candida Antarctica Lipase B (CALB) inside 2D-SiNTs (designated as CALB/2D-SiNTs), characterized by Fourier transform infrared spectroscopy (FTIR) and Raman spectroscopy (Supplementary Figs. 23–25)[46]. After assembling CALB/2D-SiNTs on yeast (designated as Yeast@CALB/2D-SiNTs), we planned a two-step reaction sequence: yeast-mediated reduction of aldehyde to benzyl alcohol followed by CALB-mediated trans-esterification in one-pot (Fig. 6d, e, Supplementary Fig. 26). A reaction mixture containing Yeast@CALB/2D-SiNTs, benzaldehyde and vinyl n-hexanoate was stirred at 30 °C to result ca. 90% yield of benzyl alcohol ester, formed by a two-step sequential reaction by yeast and CALB components (Fig. 6f). A control experiment using a physical mixture of Yeast@2D-SiNT and CALB

could produce only <25% yield of the desired ester under identical conditions, confirming the crucial role of integrated nanobiohybrid structure resulted from the silica-tiling strategy.

In Summary, we have developed rational integration of living microorganism (yeast) cell, with tailorable catalysts via a hierarchical inorganic protocellular confinement-based silica-tiling strategy. The utilization of highly anisotropic and mechanically flexible 2D-SiNTs as a functional scaffold has enabled a precise and conformal coating (only <10 nm thick) of individual cell, facilitating the direct interfacing of well-preserved catalytic modules with the cell membrane. Functional evaluations of these nanobiohybrids showcased robust catalytic performances by combining diverse synthetic organic transformations with natural biochemical steps. Optimally protected and re-generable cell-based catalysts can be vital for desired chemical synthesis due to their moderate resilience, allowing survival under various reaction conditions and simultaneous propogation through natural cell-division process for repeated use multiple times. Unlike thick protective shells, thin and dynamic tiles minimally compromise the required molecular diffusion during catalysis and maintain the optimum metabolism and viability which is crucial for whole cell biocatalysis. Combining living cells with heterogeneous metal-based catalysts can add versatility of synthetic catalysts and opportunity to perform multiple chemobiotic reaction steps in single pot, increasing the efficiency and sustainability. However, practical suitability of these catalysts for scalable reactions still remains to be assessed and optimized for targeted industrial application. By rationally merging advanced nanocatalyst design, self-assembly process with the natural cell biocatalyst, our work establishes a paradigm for developing versatile nanobiohybrid catalytic platforms. Our strategy holds immense potential for diverse sustainable chemical synthesis methodologies and advancing applications in pharmaceuticals, green chemistry, and industrial fine-chemical synthesis. The comprehensive understanding gained from this study paves the way for future designs of programmable and multifunctional living cell-based catalysts.

## Methods

### Synthesis of negative and positive charged 2D-SiNTs
First, silica encapsulation of single-layered (1 nm thick) NiCo layered double hydroxide (SL-LDH) was performed following a reverse microemulsion procedure using tetraethyl orthosilicate (TEOS) and N-[3-(trimethoxysilyl)propyl]ethylenediamine (TMSD), affording LDH@SiO$_2$ nanosheets (details in Supplementary Information). LDH@SiO$_2$ (5 mg) was treated with the 3 M HCl (5 mL) under stirring overnight at room temperature to etch-out LDH core and obtain internally amine-functionalized hollow 2D silica bilayer nanosheets (negatively charged 2D-SiNTs) with a 1 nm gap. After the reaction, precipitates were obtained by centrifugation and washed with ethanol and DI. To change the surface charge of 2D-SiNTs to positive, we dispersed 1 mg of as synthesized 2D-SiNTs in 1 ml of ethanol containing 40 μL of TMSD, and vortexed the mixture for 2 h. Finally, positively charged 2D-SiNTs were washed with DI and ethanol and stored in ethanol for further use.

### Synthesis of AuPt/2D-SiNTs
Initially, 0.5 mL of an aqueous suspension of pre-synthesized negatively charged 2D-SiNTs (2 mg mL$^{-1}$) was combined with 0.5 mL of a freshly prepared gold precursor solution (HAuCl$_4$• x H$_2$O, 15 mM). The reaction mixture was stirred for 2 h at room temperature. Subsequently, the mixture was washed twice with DI. Next, 0.2 mL of a sodium borohydride solution (NaBH$_4$, 100 mM) was added to the above solution. The mixture was stirred for 10 min at room temperature and thoroughly washed with ethanol and DI to result 2D-SiNTs modified by Au-seeds at the hollow interior. 0.5 mL of 2D-SiNTs containing Au-seeds (2 mg mL$^{-1}$) was combined with 0.5 mL of a freshly prepared platinum precursor solution (Na$_2$PtCl$_4$• x H$_2$O, 10 mM). The

reaction mixture was stirred for 30 min at room temperature. Subsequently, 0.5 mL of an ascorbic acid solution (AA, 20 mM) was quickly added to the above solution. The entire reaction mixture was then kept on a preheated oil bath (70 °C) under stirring for 10 min. Finally, AuPt/2D-SiNTs as the resulting black-colored material was isolated from the reaction solution via centrifugation, washed with ethanol and DI.

### Synthesis of Pd/2D-SiNTs
Initially, 0.5 mL of an aqueous suspension of pre-synthesized negatively charged 2D-SiNTs (2 mg mL$^{-1}$) was combined with 0.5 mL of a freshly prepared palladium precursor solution (Na$_2$PdCl$_4$•3H$_2$O, 15 mM). The reaction mixture was stirred for 2 h at room temperature. Subsequently, the mixture was washed twice with DI. Next, 0.2 mL of a sodium borohydride solution (NaBH$_4$, 100 mM) was added to the above solution. The mixture was stirred for 10 min at room temperature. The resulting Pd/2D-SiNTs was then thoroughly washed with ethanol and DI.

### Synthesis of CALB/2D-SiNTs
First, to facilitate electrostatic attachment between SL-LDH (0.4 mL, 12.5 mg mL$^{-1}$) and CALB enzyme (1 mg mL$^{-1}$), both were suspended in DI at pH 7.0 for 30 min. After isolating SL-LDH loaded with CALB, silica encapsulation was performed using a sol-gel method (details in Supplementary Information). After reaction, CALB/2D-SiNTs nanosheets were collected by centrifugation and washed with ethanol and DI for further use.

### Silica-tiling process to synthesize Yeast@2D-SiNTs
1 mg of yeast (4 × 10$^8$ cells mL$^{-1}$), obtained after centrifugation of a 5 mL culture, was mixed with 0.3 mg of positively charged 2D-SiNTs suspension in PBS for 5 min at room temperature. Subsequently, the encapsulated yeast cells were collected by centrifugation, washed twice with PBS (2000 rpm, 3 min) to remove excess particles, and modified cells were resuspended in 1 mL of PBS for further use. In order to get different nanobiohybrids, Yeast@X/2D-SiNTs (X = AuPt, Pd or CalB) were synthesized following the same protocol by replacing 2D-SiNTs with Au/Pt/2D-SiNTs, Pd/2D-SiNTs and CalB/2D-SiNTs. For plasmonic functionalization, 1 mL of citrate-capped AuNPs (5.0 × 10$^{13}$ pariclels mL$^{-1}$) were mixed with Yeast@Pd/2D-SiNTs for 30 min. Yeast@Pd/2D-SiNTs homogeneously decorated by AuNPs were isolated by centrifugation for 3 min at 2000 rpm and washing twice with DI.

### Bio-reduction of methyl benzoylformate by Yeast@2D-SiNTs
A suspension of Yeast@2D-SiNTs (4 × 10$^{10}$ cells mL$^{-1}$) was gently shaken at 30 °C for 30 min. Methyl benzoylformate (0.03 mmol) was combined with 2-propanol (80 μL), and the mixture was added to the Yeast@2D-SiNTs suspension and stirred for 6 h at 30 °C. After reaction, Yeast@2D-SiNTs was separated by centrifugation and the aqueous layer was saturated with sodium chloride and extracted with ethyl acetate (3 × 20 mL). The organic layer was dehydrated with Na$_2$SO$_4$, filtered, and concentrated under reduced pressure to obtain the reduced product (90% yield).

### Bio-reduction combined with NADH-regeneration using Yeast@AuPt/2D-SiNTs
A suspension of Yeast@AuPt-SiNTs (4 × 10$^{10}$ cells L$^{-1}$) in PBS was gently shaken at 30 °C for 30 min. NAD$^+$ (1 mmol) and triethanolamine (100 mmol) were added to the suspension. Methyl benzoylformate (0.03 mmol) was combined with 2-propanol (80 μL), and the mixture was added to the reaction suspension and stirred for 6 h at 30 °C. After reaction, Yeast@AuPt/2D-SiNTs was separated by centrifugation and the aqueous layer was saturated with sodium chloride and extracted with ethyl acetate (3 × 20 mL). The organic layer was dehydrated with Na$_2$SO$_4$, filtered, and concentrated under reduced pressure to obtain the reduced product (99% yield).

## Bio-reduction combined with LSPR-induced cross-coupling using Yeast@Pd/2D-SiNTs@Au

A suspension of Yeast@Pd/2D-SiNTs@Au ($4 \times 10^{10}$ cells mL$^{-1}$) in PBS was gently shaken at 30 °C for 30 min. To this suspension, phenylboronic acid (0.25 mmol), 4-iodo-benzaldehyde (0.25 mmol) and Na$_2$CO$_3$ (0.02 mmol) with 2-propanol (80 μL), were added and the whole solution was irradiated under 632 nm laser (0.3 W/cm$^2$) with constant stirring to initiate the reaction. At each time-interval, 20 μL portion was withdrawn and after centrifugation then extracted with 180 μL ethyl acetate three times. The ethyl acetate phase was dried with anhydrous magnesium sulfate. The organic layer was concentrated under reduced pressure and the product was analyzed using $^1$H NMR.

## Bio-reduction combined with enzymetic esterification using Yeast@CALB/2D-SiNTs

A suspension of Yeast@CALB/2D-SiNTs ($4 \times 10^{10}$ cells mL$^{-1}$) in PBS was gently shaken at 30 °C for 30 min. To the suspension, 17 μL vinyl n-hexanoate and 10.1 μL benzaldehyde with 2-propanol (80 μL) were added to initiate the reaction. At each time interval, 20 μL portion was withdrawn and after centrifugation, extracted with 180 μL ethyl acetate three times. The ethyl acetate phase was dried with anhydrous magnesium sulfate and concentrated under reduced pressure to yield the product for $^1$H NMR.

### Reporting summary

Further information on research design is available in the Nature Portfolio Reporting Summary linked to this article.

## Data availability

The authors declare that all the data supporting the findings of this study are available within the article and Supplementary Information files, and also are available from the authors upon request. Source data are provided as a Source Data file. Source data are provided with this paper.

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

## Acknowledgements
This work was supported by the Basic Science Research Program through the National Research Foundation of Korea (NRF) funded by the Ministry of Science, ICT & Future Planning (MSIP) [Grants NRF-2016R1A3B1907559 (I.S.L.) and RS-2023-00243584 (A.K.).

## Author contributions
A. Kumar and I. S. Lee conceived and supervised the research. J. Oh and N. Kumari synthesized the materials and conducted catalytic experiments with partial contribution from D. Kim. The manuscript was written through contributions of all authors.

## Competing interests
The authors declare no competing interests.
