## [Peer Review File · Nature Communications]

Ultrathin Silica-Tiling on Living Cells for Chemobiotic CatalysisREVIEWER COMMENTS

Reviewer #1 (Remarks to the Author):

In this Communication, the authors adapt previously reported hollow, cationic ~2D silica-based tiles to create conformal coatings on yeast directed by positive electrostatic interactions. The tiles can be loaded with various nanoparticles, Au, AuPt, Pd, etc. to develop catalytic properties based on combining [AuPt]-catalyzed NADH regeneration or LSPR-induced [Pd]-catalyzed C-C cross-coupling or Lipase-catalyzed esterification reactions – with the natural ketoreductase activity of the yeast cell. The apparently moderate electrostatic tile/yeast interfacial interactions allow cellular propagation and possibly re-healing of the tiled exoskeleton. Tiling is reversible returning the cells to their native state according to propagation behavior. Overall the paper provides a new means of cellular encapsulation that may be of interest to the nano/bio community. There are some weaknesses of the work as reported that if strengthened could warrant publication in Nature communications:

1. In the Introduction and figure schematics, the authors make some comparisons to previous strategies for cellular encapsulation. It should be noted that this topic was recently reviewed in “Bioinspired Cell Silicification: From Extracellular to Intracellular”, *J. Am. Chem. Soc.* 2021, 143, 17, 6305–6322, 2021; <https://pubs.acs.org/doi/abs/10.1021/jacs.1c00814>, (which should be cited) and where it was pointed out that the mammalian cell silicification process developed by Kaehr et al. (Kaehr, B. et al. Cellular complexity captured in durable silica biocomposites. *Proc. Natl Acad. Sci. USA* 109, 17336–17341 (2012)) results in silica coatings that are less than 10-nm thick. This was also pointed out in your reference 12. (*ACS Nano* 16, 2164–2175 (2022) and in Guo, J., De May, H., Franco, S. et al. Cancer vaccines from cryogenically silicified tumour cells functionalized with pathogen-associated molecular patterns. *Nat Biomed Eng* 6, 19–31 (2022). <https://doi.org/10.1038/s41551-021-00795-w>). The description in your article is wrong and misleading as to the potential advantages of your approach.

2. There are further comparisons made with your ref. 30. “SupraCells: living mammalian cells protected within functional modular nanoparticle-based exoskeletons”, Zhu et. Al,

Advanced Materials, 2019, where tannic acid coordination chemistry was used to reversibly bind various nanoparticles to the surfaces of living mammalian cells resulting in enhanced protection from external stressors like UV, osmotic stress, pH, and enzymatic attack. The Zhu et al. approach was much simpler than yours and amenable to much more fragile mammalian cells. To show the benefits of your approach, you should also assess the protective characteristics of your tiled cells. Further “supracells” or other NP modified cells should be used as controls in all your catalytic experiments to prove there is a benefit of your approach.

3. Figure 1 D shows schematics of various proposed NP modified cells. These schematics should be verified through high resolution TEM of ultra-microtomed or better cryomicrotomed samples. Current TEM does not provide sufficient resolution to verify the proposed structures.

4. The practical applications/implications of your tiled cells is not apparent. How would they be used? The fact that they can propagate is interesting but why would it be important? Provide an example.

5. You do not determine the diffusive characteristics of your tiles. You need to provide a molecular weight cut off or other measure of the effective pore size and the diffusive characteristics. This is important as encapsulation of yeast in thin silica shells has been shown to result in genetic reprogramming of the cell as reported in: Fazal, Z.; et al. “Three-Dimensional Encapsulation of *Saccharomyces cerevisiae* in Silicate Matrices Creates Distinct Metabolic States as Revealed by Gene Chip Analysis. ACS Nano 2017, 11, 3560–3575.

Reviewed by C. Jeffrey Brinker

Distinguished and Regents Professor Emeritus, Departments of Chemical and Biological Engineering and Molecular Genetics and Microbiology, UNM; Member of UNM Comprehensive Cancer Center, UNM; Fellow Emeritus Sandia National Laboratories, Distinguished Affiliate Scientist Emeritus, Sandia/Los Alamos National Laboratories Center for Integrated Nanostructures (CINT). Member: National Academy of Engineering, National Academy of Inventors, American Academy of Arts and Science, and National Academy of

Sciences, Associate Editor ACS Nano

<https://brinkerlab.unm.edu/>

Reviewer #2 (Remarks to the Author):

This study reported the integration of 2D silica sheets as nanocoatings on living yeast cells. By functionalizing the silica sheets with metal NPs exogeneous enzymes, the hybrid yeasts can perform unconventional chemobiotic reactions. Although various synthetic nanomaterials have been integrated with living cells, including silica, the present study focuses on the versatile conformal coating process and the coating enabled several unique chemobiocatalysis reactions. Overall this study is nicely presented, but several key issues should be addressed.

1. Metal NPs leaching and 2D silica stability after the chemobiotic catalysis, as well as the viability of the hybrid yeast should be studied.
2. The exploration and optimization of the underlying coupling mechanism between the synthetic catalysts and the biocatalysis process are insufficiently detailed. An in-depth analysis, such as investigating the effect of varying metal nanoparticle concentrations on the rate of chemobiotic reactions, would provide valuable insights into optimizing these hybrid systems.
3. The rationale behind ensuring that the nanocoatings do not impact cell division and proliferation, especially considering that the demonstrated chemobiotic reactions occur within a short timeframe (less than 10 hours), remains unclear. Clarification on the significance of preserving these cellular processes in the context of the conducted reactions would enhance the study's comprehensiveness.

Reviewer #3 (Remarks to the Author):

Single cells nanofunctionalization is a world focus which can play an invaluable role to better understanding, manipulation and utilization of living cells. In this work, Lee and coauthors designed a highly anisotropic and mechanically flexible 2D-SiNTs for nanofunctionalizing yeast cells by electrostatic interactions with showing well catalytic performances. By a complete system work, that I think a major revision should be made before it can be

published on Nat. Commun.

My comments are as follows:

1. The author has emphasized that it is rational integration of materials on yeast cell surface, which is different from other methods with random self-assembly. In fact, 2D SiNTs are attracted by static electricity and then randomly coat on cells without an ordered structure. Please explain that why it is called rational integration.

2. For the AuPt/2D-SiNTs, what is the form of AuPt in catalysts, and is it a simple mixture or an alloy? As we all know, Au and Pt are both common catalysts for catalytic hydrogenation reduction. So, dose the AuPt involved in the reaction of reduction of ketoester (methyl benzoylformate) to the chiral hydroxyl ester (methyl 2-hydroxy-2-phenyl acetate)? The author should evaluate their catalytic performance by provide the catalytic performance of AuPt/2D-SiNTs as control.

3. In the recyclability test of Yeast@AuPt/2D-SiNTs for methyl (Figure 5g), the author should provide the SEM or TEM data of cells before and after catalysis to show the structure stability of this hybrid.

4. The use of microbial cells for biocatalysis is often very complex due to factors such as microbial proliferation and liquid environment. Therefore, the results of cell catalysis often have significant fluctuations. However, there are no probabilistic reliability analysis for the data in figure 6 c and f. The author should provide Student t-test of these data.

5. More references should be added in the introduction to learn more about biocatalysis and smart cell encapsulation strategy. (Please check Nat. Rev. Mater., doi.org/10.1038/s41578-021-00350-8; Trend. Biotech., doi.org/10.1016/j.tibtech.2022.01.012; Coordin. Chem. Rev., doi.org/10.1016/j.ccr.2023.215471; Chem. Soc. Rev., doi.org/10.1039/d3cs00369h).

Point-by-Point Redressal to the Reviewers' Comments

Reviewer #1

In this Communication, the authors adapt previously reported hollow, cationic ~2D silica-based tiles to create conformal coatings on yeast directed by positive electrostatic interactions. The tiles can be loaded with various nanoparticles, Au, AuPt, Pd, etc. to develop catalytic properties based on combining [AuPt]-catalyzed NADH regeneration or LSPR-induced [Pd]-catalyzed C-C cross-coupling or Lipase-catalyzed esterification reactions – with the natural ketoreductase activity of the yeast cell. The apparently moderate electrostatic tile/yeast interfacial interactions allow cellular propagation and possibly re-healing of the tiled exoskeleton. Tiling is reversible returning the cells to their native state according to propagation behavior. Overall the paper provides a new means of cellular encapsulation that may be of interest to the nano/bio community. There are some weaknesses of the work as reported that if strengthened could warrant publication in Nature communications:

1. In the Introduction and figure schematics, the authors make some comparisons to previous strategies for cellular encapsulation. It should be noted that this topic was recently reviewed in “Bioinspired Cell Silicification: From Extracellular to Intracellular”, J. Am. Chem. Soc. 2021, 143, 17, 6305–6322, 2021; <https://pubs.acs.org/doi/abs/10.1021/jacs.1c00814>, (which should be cited) and where it was pointed out that the mammalian cell silicification process developed by Kaehr et al. (Kaehr, B. et al. Cellular complexity captured in durable silica biocomposites. Proc. Natl Acad. Sci. USA 109, 17336–17341 (2012)) results in silica coatings that are less than 10-nm thick. This was also pointed out in your reference 12. (ACS Nano 16, 2164–2175 (2022) and in Guo, J., De May, H., Franco, S. et al. Cancer vaccines from cryogenically silicified tumour cells functionalized with pathogen-associated molecular patterns. Nat Biomed Eng 6, 19–31 (2022). <https://doi.org/10.1038/s41551-021-00795-w>). The description in your article is wrong and misleading as to the potential advantages of your approach.

Response: Following the reviewer's point, we have modified the Introduction (Page 1, line 9-12) and accordingly added the new references (ref# 14-18) in the revised manuscript.

2. There are further comparisons made with your ref. 30. “SupraCells: living mammalian cells protected within functional modular nanoparticle-based exoskeletons”, Zhu et. Al, *Advanced Materials*, 2019, where tannic acid coordination chemistry was used to reversibly bind various nanoparticles to the surfaces of living mammalian cells resulting in enhanced protection from external stressors like UV, osmotic stress, pH, and enzymatic attack. The Zhu et al. approach was much simpler than yours and amenable to much more fragile mammalian cells. To show the benefits of your approach, you should also assess the protective characteristics of your tiled cells. Further “supracells” or other NP modified cells should be used as controls in all your catalytic experiments to prove there is a benefit of your approach.

Response: As per reviewer’s suggestion, we have assessed the protective characteristics of the *tiled* cells against various stressors by estimating the cell-viabilities after exposure to different conditions (details in the revised Supplementary Information): organic solvent (2-propanol), temperature (60 °C), physical stirring (14000 rpm, rotating magnetic bead), sonication (40 kHz, 300 W), osmotic pressure (PBS X5), ROS (3 mM H₂O₂), UV light (254 nm, 4W) exposure, acidity (pH 4.0), enzymatic attack (lyticase, 200 unit) and toxic metal nanoparticles (*ca.* 20 nm AgNPs). Under identical conditions, cell viabilities of native yeast cells and Yeast@2D-SiNTs were reasonably comparable or slightly better in the latter case. No dramatic influence of *silica-tiling* on cell viability against strong physical forces (stirring, sonication and pressure) was observed. Also, small size molecular stressors (organic solvent, acid and radicals) appeared to interact with native Yeast and Yeast@2D-SiNTs in similar way. However, *silica-tiling* significantly suppressed the cell death upon exposure to the cell-wall degrading *Lyticase* enzyme as compared to the native yeast, possibly, due to the restricted interaction of large-size enzymes with the cell-membrane. It is to be noted that entry of few enzymes across rarely available *inter-tile* junctions can’t be fully suppressed. Such results are expected due to the ultrathin self-assembled morphology of microporous bilayer silica nanosized units, which may provide only moderate optimal protection to further employ them in catalytic reactions. We have added the relevant discussion in the revised manuscript (Page 8, line 3-15) and Supplementary Information (Figure S7).

Figure S7. Resistance test of Yeast@2D-SiNTs under various stressors. Cell viability comparisons of Yeast and Yeast@2D-SiNTs in normal conditions, alcohol (2-propanol), temperature (60 °C), physical stress (stirring 1400 RPM), sonication (40 kHz, 300 W), osmotic pressure (PBS X5), pH (pH 4.0), ROS (H₂O₂ 3mM), UV (254 nm 4 W), Lyticase (*Arthrobacter luteus*, 200 unit), toxic NPs (Ag NPs 20 nm, 20 µg/mL). Error bars have been obtained by 3 independent experiments in each case.

Further, following the Reviewer's suggestion, we prepared a control catalyst – hollow silica nanoshell modified with AuPt NCs (AuPt/*h*-SiO₂) and assembled with the yeast cells (designated as Yeast@AuPt/*h*-SiO₂). As shown in Figure S17, positively charged spherical shape NPs clustered randomly on to the yeast cell membrane. Using Yeast@AuPt/*h*-SiO₂ as control catalyst, we performed the reduction of ketoester (methyl benzoylformate) to the hydroxyl ester (methyl 2-hydroxy-2-phenyl acetate) in the presence of exogenous β-NAD⁺, resulting *ca.* 72 % yield of the final product (Figure S17). Under identical conditions, Yeast@AuPt/2D-SiNTs resulted *ca.* 95% product yield (Figure 5 in the manuscript). Better catalytic performance of the *tiled* yeast catalyst may be due to the intimate monolayer interfacing of AuPt/2D-SiNTs catalysts on cell-surface, cooperating more effectively with the cell's biocatalysis. We have added the data and the discussion in the revised manuscript (Page 10, line 13-15) and SI (Figure S17).

Figure S17. Characterization and catalysis test of Yeast@AuPt/h-SiO₂ as control study. (a-b) TEM, HAADF-STEM and EDS-based elemental mapping of AuPt/h-SiO₂ NPs. (c) SEM image of Yeast@AuPt/h-SiO₂ showing random clustering of AuPt/h-SiO₂ on cell surface. (d) Reduction of methyl benzoylformate to the methyl 2-hydroxy-2-phenyl acetate in the presence of exogenous β -NAD⁺ using Yeast@AuPt/h-SiO₂. Error bars have been obtained by 3 independent experiments in each case.

3. Figure 1 D shows schematics of various proposed NP modified cells. These schematics should be verified through high resolution TEM of ultra-microtomed or better cryomicrotomed samples. Current TEM does not provide sufficient resolution to verify the proposed structures.

Response: As pointed out by the reviewer, we have conducted higher resolution Bio-TEM imaging of ultra-microtomed samples. As shown in the TEM images, 2D-SiNTs modified with different metals can be clearly visualized interfacing with the Yeast cell wall, supporting our schematics in the Figure 1D. It is to be noted that Bio-TEM sample preparation protocol required multi-step treatment with different chemicals and resin and finally cross-sectioning the cells in the dried state. Despite of the harsh-treatments, different *tiled* cells maintained their structures reasonably well with slight disturbance due to the unavoidable sample processing conditions. We have added this data in the revised manuscript (Figure 5) and SI (Figure S12).

Figure S12. TEM images showing detailed characterization of different components of Yeast nanobiohybrid catalysts. (a,c) TEM images of AuPt/2D-SiNT and Pd/2D-SiNT@Au single-sheet in standing orientation. (b,d,e) Bio-TEM images of different cells in ultra-microtomed samples.

4. *The practical applications/implications of your tiled cells is not apparent. How would they be used? The fact that they can propagate is interesting but why would it be important? Provide an example.*

Response: Optimally protected and re-generable cell-based catalysts can be vital for industrial chemical synthesis due to their moderate resilience, allowing survival under various reaction conditions and simultaneous propagation through natural cell-division process for repeated use multiple times. Unlike thick protective shells, thin and dynamic tiles minimally compromise the molecular diffusion during catalysis and maintain the optimum metabolism and viability which is crucial for whole cell biocatalysis. Combining living yeast cells with heterogeneous metal-based catalysts further adds versatility of synthetic

catalysts and opportunity to perform multiple chemobiotic reaction steps in single pot, further, increasing the efficiency and sustainability. Unlike enzymes, whole yeast cell based biohybrids offer stability, easy handling, and natural cofactors, eliminating costly purification steps and reducing waste. We have demonstrated the preliminary implications of tiled yeast cells for combining different kinds of chemobiotic sequential reactions in a single-pot manner. However, practical suitability of these catalysts for scalable reactions still remains to be assessed and optimized for targeted industrial application.

We have added this important point suggested by the reviewer in revised Conclusion part (Page 12, line 10-19).

5. You do not determine the diffusive characteristics of your tiles. You need to provide a molecular weight cut off or other measure of the effective pore size and the diffusive characteristics. This is important as encapsulation of yeast in thin silica shells has been shown to result in genetic reprogramming of the cell as reported in: Fazal, Z.; et al. "Three-Dimensional Encapsulation of Saccharomyces cerevisiae in Silicate Matrices Creates Distinct Metabolic States as Revealed by Gene Chip Analysis. ACS Nano 2017, 11, 3560–3575.

Response: As suggested by the reviewer, we attempted to determine diffusive characteristics of *silica-tiles*-on-yeast surface. We treated Yeast@2D-SiNTs with 2-(7-Nitro-2,1,3-benzoxadiazol-4-yl)-D-glucosamine (2-NBDG), a fluorescent tracer used for monitoring glucose internalization into living cells. Confocal microscopy detected emergence of intracellular green fluorescence due to the facile 2-NBDG diffusion and cellular uptake in Yeast@2D-SiNTs, similar to the native yeast cells (Figure S8). Whereas, membrane-impermeable dye [Dil C₁₈(3)] molecules (red fluorescence) accumulated on the cell wall in both the cases. Considering the small molecular size (< 1 nm), diffusion of 2D-NBDG is consistent with the expected microporosity of amorphous silica 2D-SiNTs enclosure. Further, the fact that *silica-tiling* significantly suppressed the cell-death upon exposure to the cell-wall degrading *Lyticase* enzyme as compared to the native yeast (Figure S8), indicated the restricted entry of large-size (MW=29~55 kDa., size 5-10 nm) enzymes across the silica-tiles. It is to be noted that entry of few enzymes across rarely available *inter-tile* junctions can't be fully suppressed. Overall, diffusion of small molecules for prospective catalytic reactions across *silica-tiles* is not restricted and is very similar to the native yeast. However, entry of large size biomolecules (such as proteins) can be partially restricted by the *silica-tiles*. We have added the brief discussion about the diffusive characteristics of silica tiles in the revised manuscript (Page 8, line 15-23) and Supplementary Information (Figure S8).

b Treatment with *Lyticase* enzyme

Figure S8. Diffusive characteristics of Yeast@2D-SiNTs. (a) CLSM based fluorescence images (left to right: green, red, and merged) showing facile uptake of 2-NBDG, a fluorescent tracer (green) in contrast to the membrane impermeable DiI C₁₈(3) (red), in both the cases (Yeast and Yeast@2D-SiNTs). (b) Cell viabilities of Yeast and Yeast@2D-SiNTs after treating with *Lyticase* enzyme for different times. Error bars have been obtained by 3 independent experiments in each case.

Reviewer #2

This study reported the integration of 2D silica sheets as nanocoatings on living yeast cells. By functionalizing the silica sheets with metal NPs exogenous enzymes, the hybrid yeasts can perform unconventional chemobiotic reactions. Although various synthetic nanomaterials have been integrated with living cells, including silica, the present study focuses on the versatile conformal coating process and the coating enabled several unique chemobiocatalysis reactions. Overall this study is nicely presented, but several key issues should be addressed.

1. Metal NPs leaching and 2D silica stability after the chemobiotic catalysis, as well as the viability of the hybrid yeast should be studied.

Response: To analyze the stability of hybrid catalyst, we performed Bio-TEM, ICP-AES elemental quantification and cell-viability studies on the Yeast@AuPt/2D-SiNTs isolated after using in chemobiotic catalysis reaction (NADH generation + methyl benzoylformate bio-reduction). Bio-TEM images of the fixed and sectioned cells revealed reasonably intact positioning and layered morphology of AuPt/2D-SiNTs on cell-membrane (Figure S15). Comparing the ICP-AES of the catalyst pallet (after centrifugation) and supernatant solution, before and after reaction, estimated almost no detectable leaching of Au and Pt metals during reaction (Figure S15). In addition, cell viability studies on the hybrid catalyst isolated after reaction, revealed minimal loss (< 5%) in living cell population (Figure S15). Catalyst recyclability in generating high product yields (> 90%) even after 5 cycles also corroborates with the morphological, elemental and physiological stability of the nanobio hybrid catalyst (Figure 5). We have added the discussion and corresponding data in the manuscript (Page 9, line 23 - Page 10, line 2) and Supplementary Information (Figure S15).

Figure S15. Stability test of Yeast@AuPt/2D-SiNTs. (a) Bio-TEM images of ultra-microtomed samples showing intact morphology of *tiles* after using in the catalysis reaction. (b) ICP-AES analysis to confirm leaching of metals. (c) Cell viability before and after using the catalyst in the reaction. Error bars have been obtained by 3 independent experiments in each case.

2. The exploration and optimization of the underlying coupling mechanism between the synthetic catalysts and the biocatalysis process are insufficiently detailed. An in-depth analysis, such as investigating the effect of varying metal nanoparticle concentrations on the rate of chemobiotic reactions, would provide valuable insights into optimizing these hybrid systems.

Response: As suggested by the reviewer, we varied the concentration of metal (*ca.* 3, 9 and 17 wt% Pt), the active metal catalyst for NADH regeneration, in AuPt/2D-SiNTs and used them to prepare hybrids with yeast cells (Figure S16). Upon increasing the Pt amounts, exogenous NADH generation rates were enhanced. And, in response, benzoylformate bio-reduction rates were also increased in the one-pot sequential reaction. These results indicated the effective coupling of *on-membrane* catalytic NADH generation reaction step with intracellular bio-reduction reaction step (Figure S16). Lipophilic NADH molecules produced on membrane-interfacial catalytic sites inside silica tiles, are efficiently taken up by

the encapsulated cells, enhancing the yield of intracellular bio-reduction step. This hypothesis is also supported by the fact that other chemobiotic sequential reactions (Figure 5) involving [Pd]-catalyzed C-C cross coupling (as first step) and CALB-catalyzed enzymatic esterification (as second step) were much higher when catalytic sites (in Yeast@Pd/2D-SiNT@Au and Yeast@CALB/2D-SiNTs, respectively) were directly interfacing with the cell membrane. Adding a random non-integrated mixture of catalyst and yeast cells performed poorly in overall sequential reaction probably, due to the inefficient cellular uptake resulting from the low concentration availability of intermediate molecules generated in the solution bulk. We have detailed the reaction mechanism in the revised manuscript (Page 10, line 2-9) and included the data in Supplementary Information (Figure S16).

Figure S16. Catalytic reaction test of Yeast@AuPt/2D-SiNTs with variable amounts of metals. (a) TEM images and EDS-based metal wt% data of different AuPt/2D-SiNTs. (b) NADH generation rates of different reactions using catalysts (1-3: increasing Pt amounts) based on the increase in absorbance at 360

nm in UV-vis spectrophotometry. (c) Corresponding reaction kinetics of methyl benzoylformate bio-reduction in the presence of NAD⁺, as measured by ¹H NMR.

3. The rationale behind ensuring that the nanocoatings do not impact cell division and proliferation, especially considering that the demonstrated chemobiotic reactions occur within a short timeframe (less than 10 hours), remains unclear. Clarification on the significance of preserving these cellular processes in the context of the conducted reactions would enhance the study's comprehensiveness.

Response: Although the currently demonstrated reaction times are relatively short (<10 hours) before cells reach high numbers by natural cell-division, the *tiling* strategy remains significant for several reasons in the present context. (i) High recyclability: optimally protected and re-generable cell-based catalysts can be vital for “one-pot” sequential chemical synthesis due to their moderate resilience, allowing survival under various reaction conditions and simultaneous propagation through natural cell-division process for repeated use multiple times. (ii) Facile molecular diffusion and biochemical activity: Unlike thick protective shells, thin and dynamic tiles minimally compromise the molecular diffusion during catalysis and maintain the optimum metabolism and viability which is crucial for efficient biocatalysis. We have demonstrated the preliminary implications of tiled yeast cells for combining different kinds of simple benchmark chemobiotic sequential reactions in a single-pot manner. However, practical suitability of these catalysts for more challenging scalable reactions still remains to be assessed and optimized in future. We have added the relevant discussion in the Conclusion part of the revised manuscript (Page 12, line 10-19).

Reviewer #3

Single cells nanofunctionalization is a world focus which can play an invaluable role to better understanding, manipulation and utilization of living cells. In this work, Lee and coauthors designed a highly anisotropic and mechanically flexible 2D-SiNTs for nanofunctionalizing yeast cells by electrostatic interactions with showing well catalytic performances. By a complete system work, that I think a major revision should be made before it can be published on Nat. Commun.

My comments are as follows:

1. *The author has emphasized that it is rational integration of materials on yeast cell surface, which is different from other methods with random self-assembly. In fact, 2D SiNTs are attracted by static electricity and then randomly coat on cells without an ordered structure. Please explain that why it is called rational integration.*

Response: We emphasize the *silica-tiling* strategy to be “rational” due to the following reasons:

(i) *Shape-dependent self-assembly:* Optimally thin 2D-SiNTs are flexible enough to assume the spherical surface curvature and maximize the electrostatic adhesion by slightly bending while inter-sheet face-to-face stacking is avoided by the electrostatic repulsion so that only monolayer is continuously formed through edge-to-edge supporting around curved cell-wall. Such structural flexibility and surface chemistry is logically unique in highly anisotropic 2D-SiNTs which is distinct from the other shapes and compositions (Figure 4). Although, *silica-tiling* doesn't generate crystal-like order, minimal random stacking of nanoparticles can be fully avoided, resulting homogeneously coated individual living cell.

(ii) *Adaptability allowing cell-division:* Unlike conventional *thick-continuous-static* shells – thin and dynamic *silica-tiles* easily re-adjust by opening up the gateway for daughter cell protrusion during mitosis (Figure 3). They minimally compromise the molecular diffusion during catalysis and maintain the optimum metabolism, proliferation and viability which is crucial for whole cell biocatalysis.

(iii) *Bilayer catalytic nanospace interfacing cell-membrane:* Hierarchical inorganic protocellular functional bionic jacket around individual living cell is inspired by the periplasmic lipid inter-membrane space. Similar to the *tiling-on-roof*, 2D-SiNTs self-assemble on the membrane in a lateral fashion by self-supporting each other on the edges, and establishing a secondary compartmentalization to host and safeguard molecularly accessible metal catalytic sites.

We have clarified this important point in the Introduction part of the revised manuscript (Page 4, line 5-18).

2. For the AuPt/2D-SiNTs, what is the form of AuPt in catalysts, and is it a simple mixture or an alloy? As we all know, Au and Pt are both common catalysts for catalytic hydrogenation reduction. So, dose the AuPt involved in the reaction of reduction of ketoester (methyl benzoylformate) to the chiral hydroxyl ester (methyl 2-hydroxy-2-phyenyl acetate)? The author should evaluate their catalytic performance by provide the catalytic performance of AuPt/2D-SiNTs as control.

Response: AuPt/2D-SiNTs was synthesized by a pre-deposited Au-seed mediated Pt growth selectively inside silica bilayer using ascorbic acid as a reductant. Formation of AuPt alloy in these reaction conditions is unlikely. We performed HRTEM and XRD analysis of AuPt/2D-SiNTs to conclude the presence of a simple metallic Au and Pt mixture. As per reviewer's comment, we have performed catalytic performance of AuPt/2D-SiNTs for methyl benzoylformate bio-reduction in the presence of NAD⁺, affording < 20% yield of hydroxyl ester. This control experiment suggested poor activity of the AuPt/2D-SiNTs towards keto-ester reduction without Yeast. It supports a co-operative mechanistic role of AuPt/2D-SiNTs and Yeast in NADH generation and keto-ester reduction steps, respectively. We have added the corresponding discussion and data in the manuscript (Page 10, line 16-17) and Supplementary Information (Figure S18).

Figure S18. Characterization and catalytic reaction test of AuPt/2D-SiNTs (a-b) TEM and HRTEM images of AuPt/2D-SiNTs, showing Au-seed mediated dendritic growth of Pt. (c) XRD data of AuPt/2D-SiNTs showing existence of Au and Pt simple mixture. (c) Conversion yield of methyl benzoylformate bio-reduction in the presence of NAD^+ using AuPt/2D-SiNTs (without Yeast), as measured by ^1H NMR. Error bars have been obtained by 3 independent experiments in each case.

3. In the recyclability test of Yeast@AuPt/2D-SiNTs for methyl (Figure 5g), the author should provide the SEM or TEM data of cells before and after catalysis to show the structure stability of this hybrid.

Response: To analyze the stability of hybrid catalyst, we performed Bio-TEM, ICP-AES elemental quantification and cell-viability studies on the Yeast@AuPt/2D-SiNTs isolated after using in chemobiotic catalysis reaction (NADH generation + methyl benzoylformate bio-reduction). Bio-TEM images of the fixed and sectioned cells revealed reasonably intact positioning and layered morphology of AuPt/2D-SiNTs on cell-membrane (Figure S15). Comparing the ICP-AES of the catalyst pallet (after centrifugation) and supernatant solution, before and after reaction, estimated almost no detectable leaching of Au and Pt metals

during reaction (Figure S15). In addition, cell viability studies on the hybrid catalyst isolated after reaction, revealed minimal loss (< 5%) in living cell population (Figure S15). Catalyst recyclability in generating high product yields (> 90%) even after 5 cycles also corroborates with the morphological, elemental and physiological stability of the nanobio hybrid catalyst (Figure 5). We have added the discussion and corresponding data in the manuscript (Page 7, line 23 - Page 8, line 3) and Supplementary Information (Figure S15).

Figure S15. Stability test of Yeast@AuPt/2D-SiNTs. (a) Bio-TEM images of ultra-microtomed samples showing intact morphology of *tiles* after using in the catalysis reaction. (b) ICP-AES analysis to confirm leaching of metals. (c) Cell viability before and after using the catalyst in the reaction. Error bars have been obtained by 3 independent experiments in each case.

4. The use of microbial cells for biocatalysis is often very complex due to factors such as microbial proliferation and liquid environment. Therefore, the results of cell catalysis often have significant fluctuations. However, there are no probabilistic reliability analysis for the data in figure 6 c and f. The author should provide Student t-test of these data.

Response: As per author's comment, we have performed statistical analysis in the catalysis experiments. Accordingly, we have revised the Figure 6 in the manuscript by adding statistical error bars obtained from the results of three independent experiments.

5. More references should be added in the introduction to learn more about biocatalysis and smart cell encapsulation strategy. (Please check Nat. Rev. Mater., doi.org/10.1038/s41578-021-00350-8; Trend. Biotech., doi.org/10.1016/j.tibtech.2022.01.012; Coordin. Chem. Rev., doi.org/10.1016/j.ccr.2023.215471; Chem. Soc. Rev., doi.org/10.1039/d3cs00369h).

Response: Suggested references have been added (Ref# 3-6).

REVIEWER COMMENTS

Reviewer #1 (Remarks to the Author):

The authors have responded moderately to my original comments, however there remain a few inconsistencies between the results and their descriptions. Additionally, the sentence beginning “Yeast cell, a widely-used biocatalyst, is ingeniously upgraded via highly controlled self-assembly of 2D-bilayer silica-based catalytic modules on cell surfaces, opening the avenues for diverse chemobiotic reactions:” must be edited to remove the self-proclaimed “ingeniously upgraded” language.

1. Fig 2h shows substantial tiling heterogeneity – this needs further explanation.
2. The effective pore size of the tiled coating was established to be less than about 5-nm. The size of yeast cells buds at the parent cell membrane surface has been shown to be about 1 μ m. Obviously this bud cannot protrude through a pore in the tiling – rather it must occur through the emergence and pinching off of the parent membrane. How is it that his process results in few tiles being on the daughter cells?
3. The response of the tiled cells to stressors is rather weakly characterized showing for example one level of osmotic stress, one level of pH etc. The results should be presented as viability versus the value/magnitude of the stressor (T, pH, osmotic stress etc).

Reviewer #2 (Remarks to the Author):

The authors have addressed most of my concerns. It can be accepted for publication.

Reviewer #3 (Remarks to the Author):

The authors have modified the manuscript and replied the comments properly that I think it can be accepted.

Point-by-point response to the Reviewers' comments

Reviewer #1:

The authors have responded moderately to my original comments, however there remain a few inconsistencies between the results and their descriptions. Additionally, the sentence beginning “Yeast cell, a widely-used biocatalyst, is ingeniously upgraded via highly controlled self-assembly of 2D-bilayer silica-based catalytic modules on cell surfaces, opening the avenues for diverse chemobiotic reactions:” must be edited to remove the self-proclaimed “ingeniously upgraded” language.

Response: We thank reviewer for making highly constructive comments on the manuscript. It has significantly increased the quality of our work after revision. As further pointed out by the reviewer, we have corrected the misleading statement in the revised abstract.

1. Fig 2h shows substantial tiling heterogeneity – this needs further explanation.

Response: In Figure 2h, we had acquired CLSM images of freely dispersed cells in liquid media after assembling 2D-SiNTs (labeled green) on yeast cell membrane (labeled red). Heterogeneity of green fluorescence (*tiling*) in high-mag CLSM images may be due to different reasons: (i) Ultrathin *tiling* provides much weaker and fluctuating fluorescence signals as compared to the interfacial thick cell-wall. (ii) Slight movements or adjustments of nanoscale silica *tiles* on living and slightly mobile cells (*non-fixed*) can't be fully suppressed during measurements. These heterogeneity factors may be broadly resolved and visualized in high-mag live-cell CLSM imaging. However, low-mag CLSM images (Figure S3) of crowded cells showed reasonably homogeneous presence of green 2D-SiNTs fully coated on every individual yeast cell. In addition, Bio-TEM and SEM images of fixed cells confirmed lateral monolayer assembly of 2D-SiNTs on cell-membrane. We have added this explanation in the revised manuscript (Page 6, line 12-18).

2. The effective pore size of the tiled coating was established to be less than about 5-nm. The size of yeast cells buds at the parent cell membrane surface has been shown to be about 1 μ m. Obviously this bud cannot protrude through a pore in the tiling – rather it must occur through the emergence and pinching off of the parent membrane. How is it that his process results in few tiles being on the daughter cells?

Response: Although the effective pore size of the *tiled* coating was less than 5 nm (estimate by the molecular diffusion studies), the protrusion of the daughter cell may occur through pinching off the 2D-SiNT shared edges, opening a sufficiently larger size gate to facilitate the outwardly protruding daughter

cell-growth and finally de-attachment from the mother cell. CLSM-based live-cell imaging study on the real-time cell division process of Yeast@2D-SiNTs showed budding daughter-cell protrude out of thin silica enclosure without carrying away any 2D-SiNT on their newly evolving surface. This is due to the limited availability of 2D-SiNTs remaining stably attached only on the mother cell which don't transfer to the newly emerging daughter cell. We have clarified this important point in the revised manuscript (Page 6, line 26-27; Page 6, line 1).

3. The response of the tiled cells to stressors is rather weakly characterized showing for example one level of osmotic stress, one level of pH etc. The results should be presented as viability versus the value/magnitude of the stressor (*T*, pH, osmotic stress etc).

Response: Following the Reviewer's comment, we have performed the cell viability studies with different magnitudes of stressors. We have added the data in the revised SI (Figure S7).

Figure S7. Resistance of Yeast@2D-SiNTs to endogenous and exogenous stimuli. Viability of Yeast (black) and Yeast@2D-SiNTs (grey) after treatment with different alcohol (2-propanol, 30 min) concentrations, temperatures (30 min), physical stress (stirring 1400 RPM, sonication (40 kHz 300 W), osmotic pressures (PBS, 2 h), pH (2h), ROS (H_2O_2 2 h), UV (254 nm 4W, 2 h.), Lyticase enzyme (*Arthrobacter luteus*, 2h.), Toxic NPs (Ag NPs 20 nm, 2 h).

Reviewer #2:

The authors have addressed most of my concerns. It can be accepted for publication.

Response: We thank reviewer for making highly constructive comments on the manuscript. It has significantly increased the quality of our work after revision.

Reviewer #3:

The authors have modified the manuscript and replied the comments properly that I think it can be accepted.

Response: We thank reviewer for making highly constructive comments on the manuscript. It has significantly increased the quality of our work after revision.

REVIEWERS' COMMENTS

Reviewer #1 (Remarks to the Author):

The authors have adequately responded to my comments and I feel this paper could be accepted.